# Sailing Too Close to the Wind? How Harnessing Patient Voice Can Identify Drift towards Boundaries of Acceptable Performance

**DOI:** 10.3390/healthcare12151532

**Published:** 2024-08-01

**Authors:** Siri Wiig, Catherine Jane Calderwood, Jane O’Hara

**Affiliations:** 1SHARE—Centre for Resilience in Healthcare, Faculty of Health Sciences, N-4036 Stavanger, Norway; 2Faculty of Science, University of Strathclyde, Glasgow G1 1XQ, UK; ccalderwood@doctors.org.uk; 3The Healthcare Improvement Studies (THIS) Institute, University of Cambridge, Cambridge CB1 8RN, UK; jane.o'hara@thisinstitute.cam.ac.uk

**Keywords:** deviance, drift, safety boundaries, patient voice

## Abstract

This opinion paper investigates how healthcare organizations identify and act upon different types of risk signals. These signals may generally be acknowledged, but we also often see with hindsight that they might not be because they have become a part of normal practice. Here, we detail how risk signals from patients and families should be acknowledged as system-level safety critical information and as a way of understanding and changing safety culture in healthcare. We discuss how healthcare organizations could work more proactively with patient experience data in identifying risks and improving system safety.

## 1. Introduction

Healthcare organizations unknowingly accept the increased risk of adverse events resulting from the cumulative effect of small reductions in care quality [1]. When signals are not identified and acted upon, organizations can drift towards the safety margins due to the normalization of deviance, including unacceptable staff and leadership behavior and performance [2,3]. As well as signaling problems in their own care, the experiences of patients and families can also be conceived as early warning signals of poor performance at the wider system level, either as negative single cases, or as general patterns of poor performance in the organizations. Here, we detail how these signals should be acknowledged as system-level safety critical information and as a way of understanding and changing safety culture in healthcare. We discuss how healthcare organizations could work more proactively with patient experience data in identifying risks and improving system safety. If they do not, they are most likely to encounter an organizational drift where the risk is uncontrolled.

## 2. Normalization of Deviance and Organizational Drift—What Is It?

In the safety science literature, there are numerous cases showing how signals of unsafe performance and deviance from safe work practices were identified, escalated, ignored, and eventually normalized over time. For example, staff may report near misses, unreliable work practices, inadequate staffing, failing technical equipment, or omitted safety procedures. Over time, particularly if these issues do not lead to adverse events, the suboptimal work practice continues, and the deviance becomes normal practice within the work culture. Whilst these small incremental erosions in the standards of practice might not be regarded in and of themselves as threatening safety, the many small omissions and adaptations leading to suboptimal practice can sometimes result in disaster. Outside of healthcare, this has been seen in high-profile events such as the Challenger explosion [3], the Deepwater horizon blowout [4], and other well-documented tragic safety events (e.g., Chernobyl, Costa Concordia). 

A seemingly universal truth is that in the aftermath of such major accidents, early warning signals of the likelihood of catastrophic failure are identified. Early warning signals can be conceived as small signs of system performance or properties (e.g., technical, procedural, cultural, hierarchical) that may indicate or contribute to future failure. What is evident in the aftermath of such accidents is that these signals had been ignored or overlooked and came to be accepted as an unproblematic part of everyday work. In safety science terms, this could be conceived as the systematic erosion of the safety margins, leading to the drift towards a state where safety failures are more likely. Put simply, they had operated too close to the boundaries of acceptable performance, and this was culturally accepted as part of everyday work. Without a continued systematic assessment of performance, organizations, and individuals, they cannot know when they are too close to these boundaries before it is too late [2]. In this paper, we argue that patients’ and families’ voices represent an untapped resource to support the closer monitoring of organizations’ proximity to the acceptable boundary of performance. 

## 3. Adverse Outcomes in Healthcare Resulting from Organizational Drift

Within England, there have been several inquiries arising from concerns of systematic poor care and outcomes in maternity services. These inquiries consistently demonstrate similar failures happening repeatedly, despite warnings from families and staff [5,6,7]. Other examples from healthcare are also highly relevant in understanding the normalization of deviance and drift into failure [8]. For example, several ‘whistleblowers’ had raised concerns about the Bristol cardiac pediatric surgery unit prior to its investigation in 2001, but no action was taken in response. Subsequent investigations uncovered evidence of significant organizational and cultural problems in the healthcare environment, with punitive management styles, surgeons who lacked professional insight but continued working, and data not matching their view were discarded thus resulting in 30–35 children dying between 1991 and 1995 [9]. The Mid Staffordshire and Morecambe Bay inquiries showed similar patterns and recommendations focusing on the need for organizational and cultural changes [7,10]. These examples collectively illustrate how healthcare organizations (or at least parts of organizations) were operating within a degraded mode for years due to then normalization of unacceptable behaviors and performance, and with work cultures having limited interest in monitoring signals from staff, and patients and families, indicating that something was terribly wrong. 

Understanding where an organization is operating in relation to acceptable performance boundaries is not straightforward [11] and requires information not just about ‘breaches’ of the boundary (safety incidents, formal patient complaints), but also identification of risk as well as constant enquiry. That enquiry should extend to the smaller, repeated failures that are often highlighted by the patient experience of care—failure to be treated with dignity and respect, of receiving basic levels of care—as well as the patient-reported harm events, which are formally reported to and handled by the organizations themselves, as well as a range of regulatory bodies in different healthcare systems. These repeated failures were evident within all the reports from the major inquiries, alongside parallel missed opportunities to learn and intervene that could have reduced the significant harm that resulted [7,9,10]. For example, families who gave birth at Morecambe Bay were often told that theirs was an isolated case, or that investigations had shown there was nothing that could have been carried out. There was evidence of uncaring practices, with staff showing little regard for those using their services. Sadly, many similar issues have been discovered elsewhere. Both Morecambe Bay and the Mid Staffordshire inquiries demonstrate how small, repeated failures were not considered as indicative of or relating to safety. We contend therefore that these ‘early warning signals’—poor quality care, lack of dignity and respect—might collectively have indicated the drift into unacceptable performance, which in turn lays the foundations for these organizational disasters [7,10]. Both inquiry reports provide a real sense that if these ‘early warning signals’ were taken seriously, and as an indication of drift, the serious safety failings might not have happened. This also means that many of the deaths could have been prevented if signals from patients and families (individual and aggregated) were valued and acted upon by professionals and leaders. This needs to change as we see the same trends being repeated today, decades later [6]. 

## 4. Discussion 

Safety critical industries have long acknowledged the need to identify, address, and use these types of early warning signals in a systemic approach to improve safety. Thinking about the road ahead, we contend that healthcare needs to recognize that it too is a safety critical industry and act in a more proactive way on signals from those who receive care. To understand how safe a healthcare organization is, we argue that the safety concerns and care experience of patients’ and family’s data goes beyond ‘soft intelligence’ [12] and should instead be conceived as fundamental safety critical information. In this final section, we outline the ways in which healthcare might move towards a systematic gathering and acting upon this safety critical intelligence from patient and families.

First, healthcare needs to systematize and legitimize information from patients and families as credible safety information to support their safety and care at the individual and system levels. Examples of these types of approaches include ‘safety netting’ and ‘red flag’ initiatives to manage diagnostic uncertainty in primary care [13]. Within acute care settings, a recent high-profile example within the UK is the implementation of ‘Martha’s Rule’. Martha Mills died of sepsis in 2021, aged 13, following a pancreatic injury sustained from the handlebar when she fell off her bike. Her family repeatedly expressed concern regarding her deteriorating condition and, in 2023, a coroner ruled that Martha would probably have survived if she had been moved to intensive care earlier. This new initiative aims to support both families and staff to access a rapid critical care review should they have concerns regarding the condition of a patient. It also promotes the systematic gathering of information from patients and families about their condition at least daily, using methods such as the ‘patient wellness questionnaire’ [14]. Such initiatives start to bridge the gap between individual signals of safety, which have been recognized as important for some time, and the wider system signals by integrating this as ‘business as usual’ alongside the more traditional methods of assessment, especially of a deteriorating patient (e.g., early warning systems).

Second, healthcare needs to use the available data from patients and families more systematically. Patient complaints are an established process within healthcare systems globally but are often seen as individual cases rather than a source of early warning signals. However, there is some evidence of the application of a validated, standardized approach to using patient complaints to identify ‘hot spots’ and ‘blind spots’ for patient harm within the Irish healthcare system [15]. Such approaches reconceptualize complaints as potential early warning signals.

Third, healthcare needs to systematically gather, attend to, and act on the plethora of data provided by patients and their families about the quality and safety of care outside of formal complaint systems. This was termed over a decade ago as the ‘patient experience cloud’ [16] to describe the large and ever-expanding source of data that are shared every day via the internet—and especially on social media—from people describing their experience of the quality and safety of healthcare. Importantly, there is emergent evidence that this cloud of data is associated with, and potentially predictive of, objective measures of quality including readmission rates, mortality, and infection rates [16,17]. The potential sources of these data on patient experience are enormous, and go beyond social media, extending to sites that systematically invite and curate patient and family experiences of health and social care. An example of this is Care Opinion, which is a free to access, non-profit website operating in an increasing number of health systems globally. It invites patients and families to leave feedback (anonymously if preferred) about their experience of health and social care. Using this platform, a recent study demonstrated that an automated analysis of the language used provides an outlet for reporting safety issues that may have been unnoticed or unresolved within formal channels [18]. The advances in automation of free text analysis [19] provide a potential mechanism for the systematic use of such sources by service providers, policymakers, and regulators to identify and potentially prevent organizational drift in healthcare.

Finally, to harness these concerns and experiences routinely to avoid operational failure will require resources, infrastructure, and design, as well accepting these sources of data as credible safety critical information. It is useful here to think about epistemic injustice [20]—whose voices are credible, sought out, and valued? Some studies have begun to explore this in relation to investigatory and regulatory processes following patient harm [17,21], where patients’ and families’ perspectives have been shown to differ from the healthcare professionals’ and are often devalued compared to the views of the ‘more competent’ actors. Epistemic injustice is likely therefore to be an important consideration when advocating for better monitoring and proactive use of patients’ experiences and information to prevent the breach of safety boundaries. Levers to support this systematization might include policy development and implementation, but also regulation [17]. The regulatory logic needs to move from an occupation with ‘objective’ evidence to the assessment of ‘soft’ signals [22]. This means a combination of regulatory measures to mandate the use of these types of information sources as part of work practices among the regulated organizations, as well as regulators who are interested and use this information to a stronger degree themselves as sources when monitoring system and organizational performance [17,23]. We acknowledge that this is difficult, but we believe that a careful and thoughtful approach as indicated above is needed and should be codesigned to better use these untapped sources of information.

## 5. Conclusions

Listening to and acting on the experiences of patients and families at both the individual and system levels may help reduce hierarchy and the risk of services crossing the boundary of acceptable performance; of ‘sailing too close to the wind’. As Don Berwick stated in his 2013 report on improving the safety of patients within the NHS—“*Hear the patient voice, at every level, even when that voice is a whisper*.” [24] (p. 17). However, achieving this systematically will require resources, new healthcare and regulatory infrastructure, and work culture changes.

## 6. Future Directions

The future direction in this field requires a culture where healthcare organizations work more proactively with patient experience data in identifying risks, preventing the normalization of deviance, and improving system safety using all available sources.

## Data Availability

No new data were created or analyzed in this study.

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
