# Peer review of "Sailing Too Close to the Wind? How Harnessing Patient Voice Can Identify Drift towards Boundaries of Acceptable Performance"

_healthcare, 2024, doi:10.3390/healthcare12151532_

Round 1
Reviewer 1 Report
Comments and Suggestions for Authors
I would like to thank the authors for a highly interesting and relevant opinion paper. You raise several important and pertinent perspectives within the field of patient safety and safety culture. Your analysis of how healthcare organisations identify and respond to various types of risk signals, particularly those from patients and their families, is insightful and valuable.
The abstract clearly captures the essence of your argument. You convincingly highlight how these risk signals should be regarded as system-level safety-critical information and how they can be used to understand and transform the safety culture in healthcare. Your further discussion on the proactive use of patient experience data to identify risks and improve system safety is particularly appreciated.
I have no changes to suggest, as I believe your paper is already well-prepared and contributes valuable insights to the field.
Thank you for a well-written and thought-provoking opinion paper.
Yours sincerely,
Author Response
Reviewer 1
I would like to thank the authors for a highly interesting and relevant opinion paper. You raise several important and pertinent perspectives within the field of patient safety and safety culture. Your analysis of how healthcare organisations identify and respond to various types of risk signals, particularly those from patients and their families, is insightful and valuable.
The abstract clearly captures the essence of your argument. You convincingly highlight how these risk signals should be regarded as system-level safety-critical information and how they can be used to understand and transform the safety culture in healthcare. Your further discussion on the proactive use of patient experience data to identify risks and improve system safety is particularly appreciated.
I have no changes to suggest, as I believe your paper is already well-prepared and contributes valuable insights to the field.
Thank you for a well-written and thought-provoking opinion paper.
Yours sincerely,
Authors’ response
We would like to thank you for your thorough reading and for enjoying our paper. Much appreciated.

Reviewer 2 Report
Comments and Suggestions for Authors
Very intersting and valuable paper.
Well written.
Some ideas:
Expand idea of "culture" of healthcare environment.
Might be idea to recommend concentrate on population group to which patient/family belong to ...ensure equity and some "fast wins".
Hierachical nature of healthcare an issue in many healthcare scandals
Interested to hear about "patient experience cloud" - This is a valuable resource.
Regulation, government polcy has important role to play.
Important to mention human rights ie right to health and highest standard of healthcare
Thanks for writing this paper and highlighting these issues.
Comments on the Quality of English LanguageVery clear
Well written
Author Response
- Very intersting and valuable paper.
- Well written.
- Some ideas:
- Expand idea of "culture" of healthcare environment.
- Might be idea to recommend concentrate on population group to which patient/family belong to ...ensure equity and some "fast wins".
- Hierachical nature of healthcare an issue in many healthcare scandals
- Interested to hear about "patient experience cloud" - This is a valuable resource.
- Regulation, government policy has important role to play.
- Important to mention human rights ie right to health and highest standard of healthcare
- Thanks for writing this paper and highlighting these issues.
Authors’ response
Many thanks for your suggestions. Given that this is a short commentary piece, we are limited by word count. However, we have tried to incorporate some of your insights to culture in healthcare, and regulation and policy making.
Reviewer 3 Report
Comments and Suggestions for Authors
Great opinion piece! I really like your introduction of "epistemic injustice" to this topic of patient voice. I also appreciated the different references you used to make your case.
Author Response
Reviewer 3
Great opinion piece! I really like your introduction of "epistemic injustice" to this topic of patient voice. I also appreciated the different references you used to make your case.
Authors’ response
Thank you for your positive response. Much appreciated.
Reviewer 4 Report
Comments and Suggestions for Authors
Dear Authors,
Some minor feedback is presented here for your consideration.
- maybe provide a definition of the term 'weak' or 'small' or 'early' signals and use one term throughout - is this akin to how the term is used in Resilience Engineering?
- it would be good to include something on patient experience feedback in the form of formal complaints. Certainly in the public health systems in UK and Ireland patients have a right to have their complaints responded to and the Office of Ombudsman is key in this. There has been a lot of work done on trying to understand safety from the perspective of patient complaints in the HCAT tool from the London School of Economics which has now been built into the Irish health system for complaint management - NIMS. This helps to identify hot spots for patient harm that we might not otherwise see.
https://www.lse.ac.uk/PBS/Research/Research-Articles/Improving-healthcare-experiences-with-HCAT
O’Dowd et al Paper on this in Irish system https://doi.org/10.1093/intqhc/mzac037
Just some minor typos
abstract
line 13 ..see with hindsight
Introduction
line 25 ..normalization of deviance including unacceptable behavior
line 28 ..general patterns of poor performance in organizations
Section 3
line 68 - the view of whom was discarded
line 82 ..evident across in all reports
line 85 were the only similar case
line 97 we see the same trends being repeated today
Discussion
line 121-123 this is not entirely clear. Is this a reference to Early Warning Systems? the Paediatric Early Warning System in Ireland (and possibly UK?) has built in patient or family concern as a score in this system. Are you also suggesting that something like this would work in adult EWS?
https://www.hse.ie/eng/about/who/cspd/ncps/paediatrics-neonatology/paediatric-early-warning-score/#:~:text=The%20Paediatric%20Early%20Warning%20System,care%20for%20deteriorating%20paediatric%20patients.
line 131 potential sources of this data
line 137 demonstrated that that automated
Author Response
Reviewer 4
Dear Authors,
- Some minor feedback is presented here for your consideration.
- maybe provide a definition of the term 'weak' or 'small' or 'early' signals and use one term throughout - is this akin to how the term is used in Resilience Engineering?
- it would be good to include something on patient experience feedback in the form of formal complaints. Certainly in the public health systems in UK and Ireland patients have a right to have their complaints responded to and the Office of Ombudsman is key in this. There has been a lot of work done on trying to understand safety from the perspective of patient complaints in the HCAT tool from the London School of Economics which has now been built into the Irish health system for complaint management - NIMS. This helps to identify hot spots for patient harm that we might not otherwise see.
- https://www.lse.ac.uk/PBS/Research/Research-Articles/Improving-healthcare-experiences-with-HCAT
- O’Dowd et al Paper on this in Irish system https://doi.org/10.1093/intqhc/mzac037
Authors’ response
Thank you for the comments. We have chosen to use only one term “early warning signals” and this is not linked to the terminology in particular used in Resilience Engineering. It is often used as early warning or weak signals in the general safety science literature. We have made reference to the role of patient complaints, but this is not a main theme in our article, so we have kept it short. Focus on patient complaints has been part of the picture for a long time and would require a type of paper focusing on when accidents have happened, if it should be more extensively covered in our manuscript. We aim to focus on signals before they occur in a proactive approach and that is why we don’t give it more space here. Thank you.
Just some minor typos
abstract
line 13 ..see with hindsight
Introduction
line 25 ..normalization of deviance including unacceptable behavior
line 28 ..general patterns of poor performance in organizations
Section 3
line 68 - the view of whom was discarded
line 82 ..evident across in all reports
line 85 were the only similar case
line 97 we see the same trends being repeated today
Discussion
line 121-123 this is not entirely clear. Is this a reference to Early Warning Systems? the Paediatric Early Warning System in Ireland (and possibly UK?) has built in patient or family concern as a score in this system. Are you also suggesting that something like this would work in adult EWS?
https://www.hse.ie/eng/about/who/cspd/ncps/paediatrics-neonatology/paediatric-early-warning-score/#:~:text=The%20Paediatric%20Early%20Warning%20System,care%20for%20deteriorating%20paediatric%20patients.
line 131 potential sources of this data
line 137 demonstrated that that automated
Authors response
Thank you for the typo identifications. All but one are corrected (data is usually regarded as plural, so we have kept the use of ‘these data’) and so is the discussion section in line 121-123.